# Evolutionary conservation of ubiquitin-like protein urmylation as revealed by *URM1* gene shuffle from archaea to yeast
Katharina Zupfer [1,3], Lars Kaduhr [1,3], Larissa Bessler[2], Mark Helm [2] & Raffael Schaffrath [1] ✉

Urm1 from yeast is a unique ubiquitin-like protein with dual functionality. It has been shown to operate in tRNA thiolation and protein urmylation, combining features typical of bacterial sulfur carriers and classical ubiquitin-like modifiers. Hence, in evolutionary terms, Urm1 may be placed at the crossroad of prokaryotic sulfur transfer and eukaryotic protein conjugation pathways. Prompted by Urm1-like proteins identified in Archaea, we examined Urm1 functional conservation using *URM1* gene shuffle from *Sulfolobus acidocaldarius* to *Saccharomyces cerevisiae*. We find that archaeal Urm1 conjugates to peroxiredoxin Ahp1, a bona fide urmylation target in yeast, but cannot support tRNA thiolation. Ahp1 conjugation requires sulfur transfer onto the archaeal Urm1 modifier from Uba4, the E1-like urmylation activator in yeast. Thus, thioactivation of archaeal Urm1 and urmylation-like conjugation are conserved and exchangeable processes between *Sulfolobus* and *Saccharomyces*. Our survey underlines that Urm1 likely occupies a key role in the evolution of the ubiquitin-like protein family.

Ubiquitin is an essential protein in eukaryotes[1] that conjugates to other proteins in a process known as ubiquitylation. This post-translational protein modification is well-characterized and critical for key biological processes including cell cycle control and proteostasis[2]. Ubiquitylation is spread among eukaryotes and involves a cascade of E1–E3 enzymes[3,4]. It starts with a specific E1 activating enzyme that adenylates ubiquitin at its C-terminal glycine. Subsequently, the activated ubiquitin forms a thioester with an E2 conjugating enzyme and conjugates to a target protein with assistance of a substrate-specific E3 ligase[5,6].

Genome wide searches in all domains of life have identified proteins with structural features typical of ubiquitin, collectively termed the ubiquitin-like (Ubl) protein family (Fig. 1A). Characteristic structural features of Ubl members are conserved throughout evolution and include a β-grasp fold (β-GF), composed of five β-sheets with a central α-helix, and a C-terminal di-glycine motif or more rarely, a single glycine residue (Fig. 1B)[7]. Some Ubl members (e.g., SUMO) are even expressed in inactive precursor forms, where the glycine motif must be exposed through proteolytic maturation[8]. Remarkably, bacterial members of the Ubl family (i.e., ThiS, MoaD) are known as sulfur carrier proteins (SCP) required for the synthesis of thio-cofactors (e.g. thiamine, molybdopterin)[9]. These proteins are also activated by E1-like enzymes but require subsequent sulfur incorporation at their C-termini in the form of thiocarboxylates for SCP function.

Among eukaryotic Ubl modifiers, the ubiquitin-related modifier 1 (Urm1) uniquely functions as an SCP for tRNA thiolation and urmylation, a

Ubl protein conjugation[10–13]. Like bacterial SCPs, Urm1 activity requires formation of a thiocarboxylate on its C-terminal di-glycine motif[8]. Thus, in evolutionary terms, Urm1 is likely placed at the crossroads of eukaryotic conjugation and prokaryotic sulfur transfer pathways[14].

In *Saccharomyces cerevisiae*, Urm1 thiocarboxylation is required for tRNA thiolation and urmylation functions. It involves a network of enzymes with dedicated desulfurase (Nfs1), sulfurtransferase (Tum1) and E1-like (Uba4) activities for sulfur mobilization and transfer onto Urm1 (Supplementary Fig. 1)[15]. Interestingly, Urm1 conjugation to target proteins differs from ubiquitylation as it appears to be mechanistically simpler and without the need for E2/E3 enzymes[16]. Recent studies highlight the importance of Urm1 in the maintenance of proteostasis under conditions of cellular stress by inducing phase separation[17] or protein persulfidation[16]. The latter post-translational protein thiolation was shown to be coupled to in vitro urmylation of target protein peroxiredoxin Ahp1[16,18].

Sulfur-dependence of Urm1 functions in tRNA thiolation and urmylation is conserved in eukaryotes[19,20], and proteins related to Urm1 have also been identified in Archaea[21–23]. One such Urm1-like protein from *Sulfolobus acidocaldarius* (Saci_0669) was shown to conjugate to proteins depending on an E1-like enzyme[22]. However, whether this modifier also needs sulfur activation or operates in sulfur transfer, such as tRNA thiolation, has yet to be demonstrated and could provide valuable insights into the evolution of prokaryotic and eukaryotic members of the Ubl protein family.

[1]Universität Kassel, Institut für Biologie, FG Mikrobiologie, Kassel, Germany. [2]Johannes Gutenberg Universität Mainz, Institut für Pharmazeutische und Biomedizinische Wissenschaften, Mainz, Germany. [3]These authors contributed equally: Katharina Zupfer, Lars Kaduhr. ✉e-mail: schaffrath@uni-kassel.de

**Fig. 1 | Ubiquitin-like members of the β-GF protein family. A** Ribbon diagrams of ThiS from *E. coli* (PDB: 1f0z) and Urm1 from *S. acidocaldarius* (AlphaFold[58]) are shown next to Urm1 from *S. cerevisiae* (PDB: 2qjl) and human ubiquitin (PDB: 1ubq). Highlighted are conserved β-GFs with a typical sheet of five antiparallel β-strands (marine), a central α-helix (orange) and C-terminal di-glycine motifs (red). **B** Sequence alignments of the proteins in (**A**) using T-Coffee (Expresso)[59,60] visualised by ESPript 3.0[61]. Blue framed boxes denote groups of conserved residues (red characters) while identical residues of the C-terminal di-glycine motifs are boxed in red. **C** Unrooted phylogenetic tree of ubiquitin-like β-GF proteins. Multiple sequence alignment of Urm1 from eukaryotes (*M. musculus, H. sapiens, A. thaliana,* and *D. discoideum*) and Archaea (*S. solfataricus, S. islandicus,* and *S. acidocaldarius*) as well as eukaryotic Ubl modifiers (SUMO, ATG8, ATG12, and UFM1) and sulfur carrier families (prokaryotic MoaD, ThiS, and eukaryotic MOCS2A) was trimmed (Noisy v1.5.12.1[46,47]) and used in maximum likelihood analyses (RAxML v8.2.12[48] implemented in raxmlGUI2.0.13[49]).

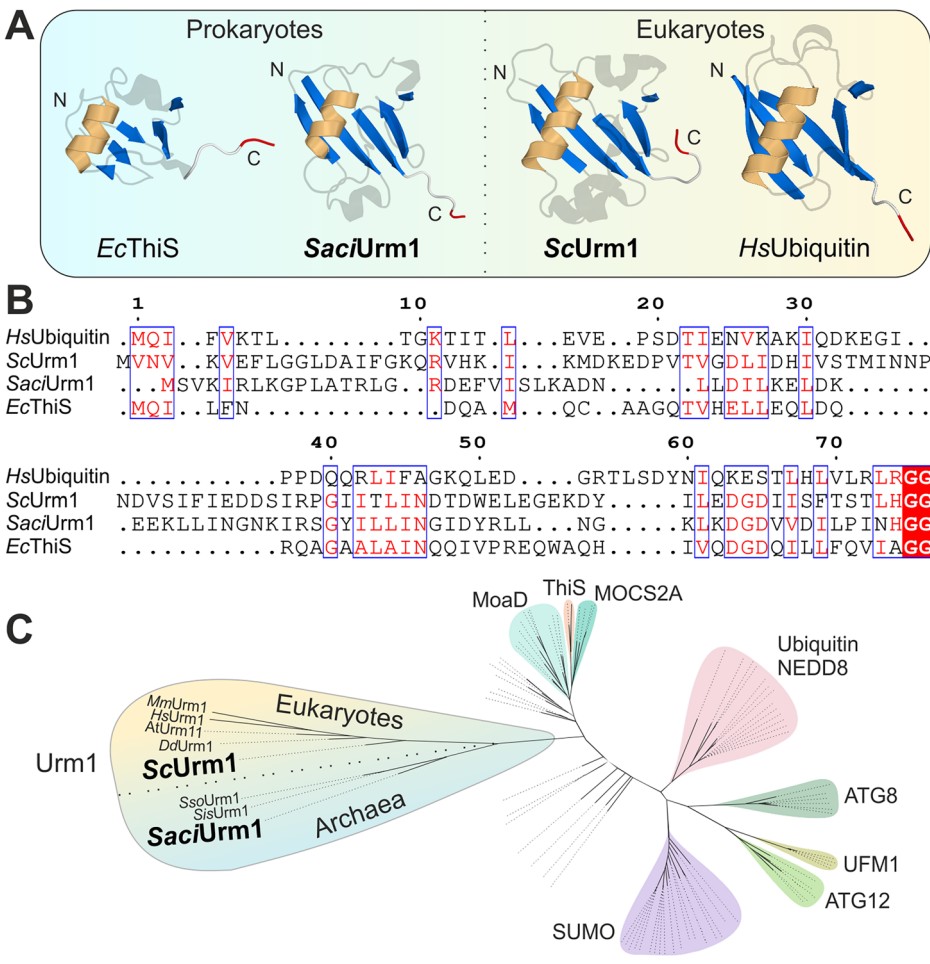

Hence, we studied Urm1 functional conservation using *URM1* gene shuffle and complementation between *S. acidocaldarius* and *S. cerevisiae*. We show that when expressed in yeast, the archaeal protein is thioactivated by Uba4 and attaches to Ahp1, analogous to *S. cerevisiae* Urm1. This shows that despite phylogenetic transition between *Sulfolobus* and *Saccharomyces*, Urm1-like conjugation has remained largely unchanged over time. Thus, the archaeal modifier qualifies as a candidate stepping-stone for studying the roots of eukaryotic members of the Ubl protein family.

## Results and discussion
### Urm1 from yeast and *Sulfolobus* cluster in a monophyletic clade
Structural and sequence comparison of ubiquitin-like β-GF proteins confirm highly conserved motifs and amino acids including the C-terminal di-glycine in all domains of life (Fig. 1A, B). Particularly, Urm1 from *S. cerevisiae* (*Sc*Urm1) and *S. acidocaldarius* (*Saci*Urm1), the proteins of our interest, show striking similarities in their sequences and overall structures (Supplementary Fig. 2). However, structural analysis alone is not sufficient to deduce information on their evolutionary relationships. Therefore, we conducted phylogenetic analyses of various Ubl proteins (i.e., ubiquitin, SUMO, NEDD8, UFM1, ATG8, ATG12, MOCS2A, MoaD, ThiS, and Urm1) that uncover a close relationship between archaeal and eukaryotic Urm1 proteins (Fig. 1C).

In our phylogenetic tree, eukaryotic Ubl proteins (SUMO, ATG8, ATG12, and UFM1) are monophyletic, while sulfur carrier families (prokaryotic MoaD, ThiS, and eukaryotic MOCS2A) are paraphyletic. Urm1 proteins occur as monophyletic group with solid bootstrap support (BS = 80; Supplementary Fig. 3). Clustering of *Saci*Urm1 (Saci_0669) in the Urm1 clade, which is the most conserved clade of the Ubl family[24,25], suggests that

they share a common ancestor and might have functional similarities (Fig. 1C and Supplementary Fig. 3).

### *Sulfolobus URM1* gene shuffle reveals urmylation-like conjugation in yeast
For complementation analysis in trans, we shuffled the *S. acidocaldarius* gene (Saci_0669) into a yeast *URM1* deletion (*urm1Δ*) strain in order to allow for expression of an HA-tagged *Saci*Urm1 variant. Based on anti-HA Western blots, the archaeal gene was robustly expressed (Supplementary Fig. 4A). To study archaeal Urm1-like conjugation in yeast, we analyzed protein extracts under conditions previously shown to detect protein urmylation[11,20]. In the presence of isopeptidase inhibitor *N*-ethylmaleimide (NEM) and based on electrophoretic mobility shift assays (EMSA), *Saci*Urm1 predominantly formed a candidate conjugate at ~40 kDa (Fig. 2A).

This shifted species is similar to the one formed between *Sc*Urm1 and yeast peroxiredoxin Ahp1 (Fig. 2B)[11,26]. Accordingly, this conjugate was not detected in extracts from an *ahp1Δ* null-mutant but became even further shifted in size (~50 kDa) when Ahp1 was c-Myc-tagged (Fig. 2A). Thus, archaeal *Saci*Urm1 attaches to Ahp1 in a fashion that is analogous to bona fide Ahp1 urmylation in yeast (Fig. 2A, B)[20]. The latter requires Lys-32 on Ahp1 as major Urm1 acceptor site and two cysteine residues (Cys-31, Cys-62). While Cys-62 is critical for peroxiredoxin activity and attachment of Urm1 at Lys-32, substitution of Cys-31 with serine shows residual activity and Ahp1 urmylation in vivo[26]. We examined whether these molecular Urm1 acceptor criteria are conserved for urmylation-like conjugation of Ahp1 by archaeal *Saci*Urm1. EMSA data obtained from *AHP1* substitution mutants clearly show that *Saci*Urm1 follows the above principles (Fig. 2C).

**Fig. 2 | *Saci*Urm1 conjugates to yeast Ahp1 in an urmylation-like fashion. A**, **B** EMSA under reducing and conjugating conditions. Protein extracts from *urm1Δ*, *urm1ΔAHP1-c-MYC* or *urm1Δahp1Δ* strains expressing *HA-SaciURM1* (**A**) or *HA-ScURM1* (**B**) were subjected to anti-HA Western blots (top panels) to detect free HA-*Saci*Urm1 (~20 kDa) or HA-*Sc*Urm1 (~23 kDa) and urmylated Ahp1 (~40 kDa) or Ahp1-c-Myc (~50 kDa) conjugates. Unconjugated Ahp1 (~20 kDa) or Ahp1-c-Myc (~25 kDa) is monitored in anti-Ahp1 Western blots (middle panels). **C** Urmylation analysis of Ahp1 mutants by EMSA. Protein extracts from *urm1Δahp1Δ* strains co-expressing *HA-SaciURM1* and indicated *AHP1* substitutions or carrying empty vector (*ev*) were subjected to anti-HA Western blots (top panel). Free forms of HA-*Saci*Urm1 (~20 kDa) and urmylated Ahp1 (~40 kDa) conjugates are marked. Free Ahp1 (~20 kDa) is detected by an anti-Ahp1 blot (middle panel). Protein loading was controlled with anti-Cdc19 Western blots (bottom panels in **A**–**C**).

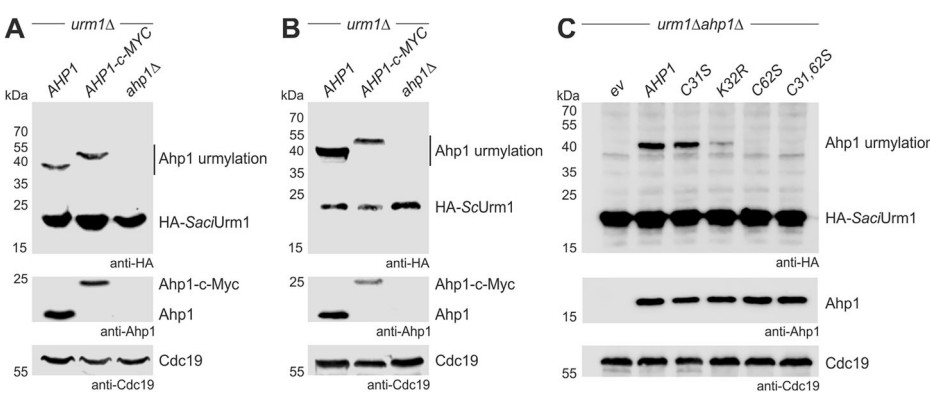

**Fig. 3 | Sulfur-dependent *Saci*Urm1 activation by Uba4 allows for Ahp1 urmylation. A** Scheme of the sulfur transfer in the yeast Urm1 pathway. Sulfur (red) flow required for Urm1 activation starts with sulfur extraction from cysteine through desulfurase Nfs1, followed by direct (Nfs1) or indirect (Tum1) sulfur transfer to the catalytic active cysteine (C397) of E1-like enzyme Uba4 that thiocarboxylates the C-terminus of Urm1. **B** Western blot analysis for sulfur dependence of *Saci*Urm1 urmylation. Protein extracts from *urm1Δuba4Δ*, *urm1Δuba4Δtum1Δ* or *urm1Δuba4Δahp1Δ* strains expressing *HA-SaciURM1* with (+) or without (-) *FLAG-UBA4* or *FLAG-UBA4-C397S* were subjected to an anti-HA Western blot (top panel) to detect free HA-*Saci*Urm1 (~20 kDa) and urmylated Ahp1 (~40 kDa) conjugates. FLAG-Uba4 or FLAG-Uba4-C397S (~55 kDa) and Ahp1 (~20 kDa) were detected by anti-FLAG and anti-Ahp1 antibodies (middle panels). Protein loading control involved an anti-Cdc19 Western blot (bottom panel).

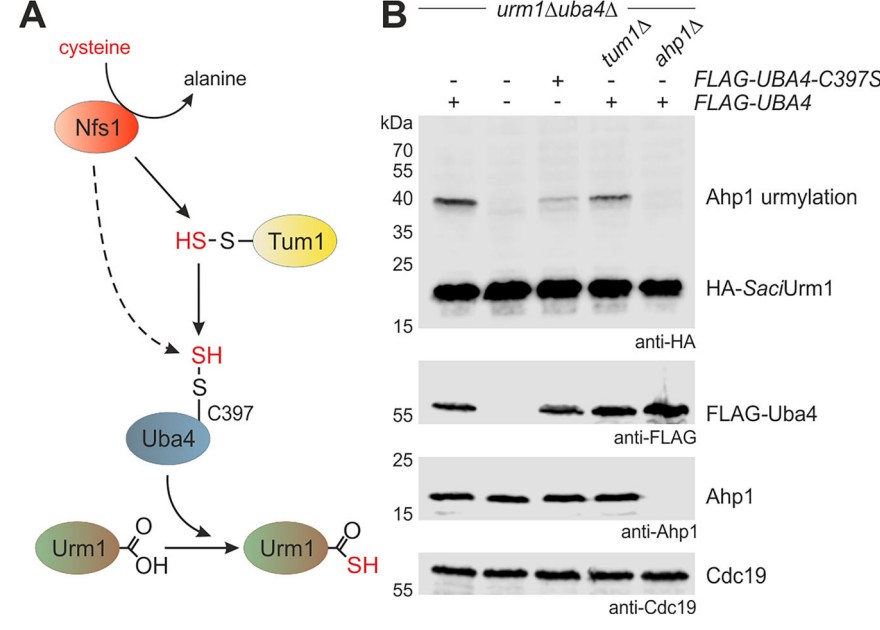

Hence, Urm1-like conjugation of Ahp1 by *Saci*Urm1 was unaltered, drastically decreased, or entirely abolished in Ahp1 variants with respective substitutions at Cys-31 (C31S), Lys-32 (K32R), or Cys-62 (C62S) alone and in tandem with Cys-31 (C31,62S) (Fig. 2C). Summing up, our data reveal that substrate specific requirements for attachment to Ahp1 are shared between an archaeal Urm1-like modifier and yeast Urm1. Thus, in evolutionary terms, Urm1-like conjugation by prokaryotic and eukaryotic modifiers of the Ubl family appears to be conserved.

**Sulfur-dependence of urmylation is conserved from archaea to yeast**

Prompted by the exchangeability between archaeal and yeast Urm1 for Ahp1 conjugation, we next examined whether sulfur dependence of Urm1 functions (i.e., urmylation, tRNA thiolation) known from yeast (Fig. 3A and Supplementary Fig. 1) is a feature conserved with *Saci*Urm1, too. Sulfur requirement for conjugation by *Saci*Urm1 was studied with sulfur transfer mutants previously shown to compromise Ahp1 urmylation in yeast[15]. The data show decreased or fully abolished Ahp1 conjugation by *Saci*Urm1 in the absence of Tum1 sulfurtransferase (*tum1Δ*) or E1-like Uba4 thioactivator (*uba4Δ*), respectively (Fig. 3B). Thus, Urm1-like conjugation to Ahp1 by *Saci*Urm1 appears to be sulfur-dependent and requires Uba4, the yeast enzyme generating the crucial Urm1 thiocarboxylate[12,13].

Since thiocarboxylation of Urm1 requires a catalytic Cys residue (C397) in the rhodanese domain of Uba4 for sulfur transfer (Fig. 3A and Supplementary Fig. 1)[13,15,27], we next studied *Saci*Urm1 conjugation to Ahp1 in yeast expressing a Uba4 mutant (C397S) defective in sulfur-dependent urmylation. Our EMSA data show a drastic effect, with Ahp1 conjugation by *Saci*Urm1 almost being abolished in the C397S mutant (Fig. 3B). This strongly suggests thioactivation of the archaeal modifier by Uba4, a scenario analogous to *Sc*Urm1 in yeast. Likewise, we found that the conserved C-terminal di-glycine motifs in *Sc*Urm1 and *Saci*Urm1 (Fig. 1B and Supplementary Fig. 2B) are required for urmylation (Supplementary Fig. 4B)[10].

**Fig. 4 | Identification of thiocarboxylated Urm1 by APM gels. A** Scheme illustrating APM usage in SDS-PAGE to diagnose Urm1 thiocarboxylation. Urm1-COOH is thio-activated at its C-terminal glycine by Uba4 resulting in Urm1-COSH. Adduct formation with thiocarboxylated Urm1 through the thiophilic character of APM results in retarded migration indicating the presence of sulfur in the thiocarboxylate (-COSH). **B, C** EMSA diagnostic for thiocarboxylation of *Saci*Urm1 (**B**) and *Sc*Urm1 (**C**) by Uba4. Protein extracts from *urm1Δuba4Δ* strains coexpressing *ScUBA4* (+) or not (-) with HA-tagged *Saci*Urm1 or HA-tagged *Sc*Urm1 or their terminal glycine deletion (ΔG) were separated by SDS-PAGE containing a layer of APM as specified (+/-). APM dependent retardation of anti-HA signals (right panels) distinguishes sulfur-free (-COOH) forms of HA-*Saci*Urm1 (**B**) or HA-*Sc*Urm1 (**C**) from thio-carboxylated (-COSH) forms.

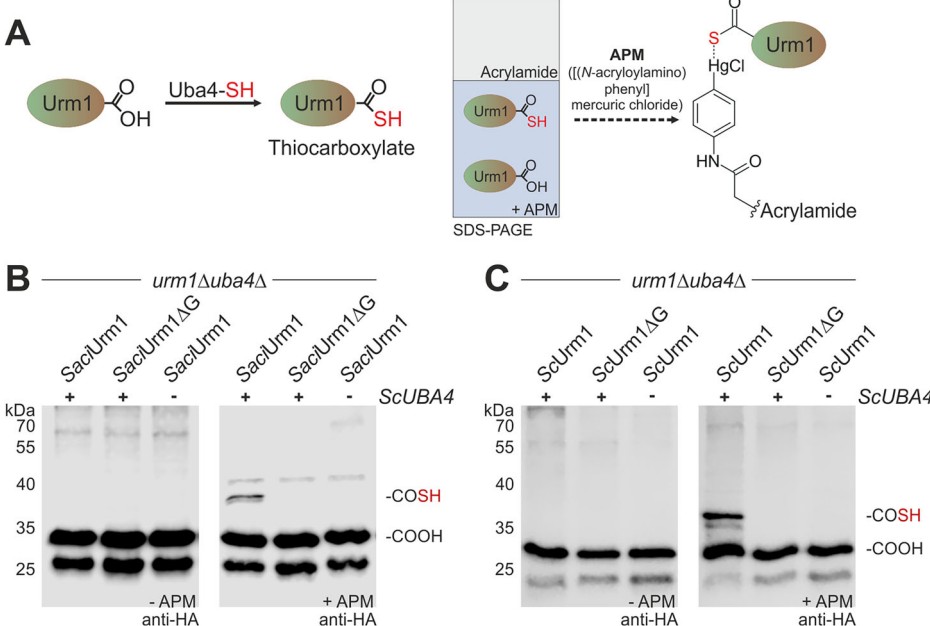

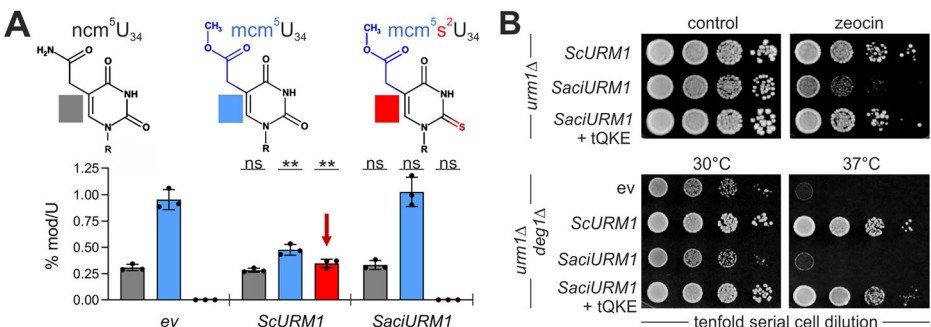

**Fig. 5 | Yeast *urm1Δ* cells expressing *SaciURM1* lack tRNA thiolation at wobble uridines (U$_{34}$). A** LC-MS/MS-based tRNA modification analysis of U$_{34}$ nucleosides (ncm$^5$U$_{34}$, mcm$^5$U$_{34}$, mcm$^5$s$^2$U$_{34}$) from *urm1Δ* strains expressing *ScURM1*, *SaciURM1* or empty vector (*ev*). The red arrow marks sulfur-dependent mcm$^5$s$^2$U$_{34}$ modification. Each measured modified nucleoside signal was normalized to total uridine content. Graphs represent the mean ± SD from n = 3 biologically independent samples; individual data points displayed as dots. Statistical significance was performed between *ev* control and *ScURM1* or *SaciURM1* expressing cells with

two-sided Student's *t*-test. *p*-values < 0.005 are indicated as ** and not significant as ns. Data values are available in Supplementary Table 4 and the Supplementary Data Excel file. **B** Phenotypic growth tests of *urm1Δ* strains expressing *ScURM1*, *SaciURM1* alone or with overexpression of tRNA$^{Gln}$, tRNA$^{Lys}$, and tRNA$^{Glu}$ (tQKE) from multi-copy plasmids on medium with DNA damaging reagent zeocin (top panel, right) or without (control). Similarly, thermo-sensitive growth was analyzed at 30 °C and 37 °C (bottom panel) in an *urm1Δ* mutant that lacks tRNA pseudouridylation gene *DEG1* (*urm1Δdeg1Δ*).

Thus, the archaeal modifier failed to form Ahp1 conjugates when its C-terminal Gly-84 was removed (ΔG) (Supplementary Fig. 4B), strongly suggesting that a thiocarboxylated C-terminus plays a crucial and universally conserved role for modifications by Urm1-like proteins. To further investigate thiocarboxylation of *Saci*Urm1 by Uba4, we used SDS-PAGE containing APM ([[(*N*-acryloylamino) phenyl]mercuric chloride)[28]. Sulfur-specific adduction of thiolated *Saci*Urm1 to APM induces mobility shifts that are detectable by Western blot (Fig. 4A)[29,30]. Our data show that APM indeed forms adducts with *Saci*Urm1 provided both functional Uba4 and a proper C-terminal di-glycine motif are present (Fig. 4B). These are requirements for APM adduction to *Saci*Urm1 that are very similar to APM retardation seen with *Sc*Urm1 (Fig. 4C).

In sum, our data indicate that thiocarboxylation and urmylation principles known from yeast also apply to the archaeal Urm1-like modifier and that sulfur-dependent performance of *Saci*Urm1 in urmylation-like conjugation is conserved from prokaryotes to eukaryotes.

## *Saci*Urm1 fails to mediate tRNA thiolation in yeast despite being thioactivated

With the confirmation of *Saci*Urm1 thiocarboxylation, we next asked whether the archaeal modifier can support sulfur transfer for tRNA thiolation. In yeast, a thiolase heterodimer (Ncs2-Ncs6) incorporates sulfur from *Sc*Urm1 into tRNA anticodons at wobble uridines (U$_{34}$). Together with other tRNA modifiers, this forms composite 5-methoxycarbonyl-methyl-2-thio (mcm$^5$s$^2$) groups on tRNA$^{Gln}$, tRNA$^{Lys}$, and tRNA$^{Glu}$ (tQKE) (Supplementary Fig. 1)[12,13,31]. Direct comparison of tRNA thiolation capacities between *Saci*Urm1 and *Sc*Urm1 by liquid chromatography-tandem mass spectrometry (LC-MS/MS) revealed that in contrast to *Sc*Urm1, mcm$^5$s$^2$U$_{34}$ formation is absent from *urm1Δ* cells expressing *Saci*Urm1 (Fig. 5A).

Similarly, expression of *Saci*Urm1, rather than *Sc*Urm1, associates with excess levels of 5-methoxycarbonyl-methyl (mcm$^5$) groups that are typical of a thiolation defect (Fig. 5A). Furthermore, LC-MS/MS revealed no impact of *Saci*Urm1 on sulfur-independent 5-carbamoyl-methyl (ncm$^5$) groups

(Fig. 5A), thus further indicating a thiolation-specific block of $Saci$Urm1 in mcm$^5$s$^2$U$_{34}$ synthesis. We also confirmed this defect in vivo by showing that $Saci$Urm1 expression cannot rescue phenotypes that result from U$_{34}$ thiolation loss in $urm1\Delta$ cells (Fig. 5B). In yeast, lack of tRNA thiolation at the wobble position U$_{34}$ does not result in a reduced abundance of hypomodified tRNAs, but has been shown to reduce the binding affinity of the hypomodified anticodon to the cognate codon in the ribosomal A-site[32,33]. The defect results in inefficient decoding and error-prone mRNA translation, which is why the growth and viability of $URM1$ gene deletion ($urm1\Delta$) strains becomes significantly compromised, particularly upon cultivation at elevated temperatures or in the presence of chemical stressors such as zeocin (Fig. 5B). Intriguingly, overexpression of hypothiolated tRNAs (i.e., tQKE) compensates for the decoding inefficiency of $urm1\Delta$ cells and partially rescues the thermo and drug sensitivity phenotypes (Fig. 5B). Unlike with $Sc$Urm1 or partial rescue by tRNA overexpression (tQKE, see above), $Saci$Urm1 expression, however, could not complement zeocin sensitivity or synthetic lethality in tandem with tRNA pseudouridylation mutant $deg1\Delta$ (Fig. 5B)[13,20,34].

Hence, our data show that despite being thiocarboxylated by Uba4 for sulfur-dependent Urm1-like conjugation (Fig. 4), $Saci$Urm1 cannot transfer the sulfur for tRNA thiolation in yeast (Fig. 5). Such incompatibility may be explained by $Saci$Urm1 failing to properly interact with the thiolase complex in yeast. In contrast to eukaryotic thiolase heterodimers, archaeal homologues rather form homodimers, indicating an evolutionary diversification from homo- to heterodimeric tRNA thiolation enzymes[30,35,36]. This could reflect an evolutionary barrier that may have resulted in functional specification of the eukaryotic thiolase complex that is unable to recognize and use the archaeal Urm1 protein as sulfur donor[36,37].

To the best of our knowledge there is no data linking $Saci$Urm1 directly to sulfur transfer associated with tRNA thiolation in $Sulfolobus$. However, archaeal tRNA thiomodifications do exist as has been recently uncovered by comprehensive LC-MS/MS profiles on tRNAs isolated from several Archaea including $S.$ $acidocaldarius$[38]. Thus, a sulfur donor function of $Saci$Urm1 for tRNA thiolation likely exists in its host organism $S.$ $acidocaldarius$ and needs further investigations. These should also include the two open reading frames from $S.$ $acidocaldarius$, predicted to encode Ubl proteins (Saci_0952 and Saci_1652) (Supplementary Fig. 3). They may represent potential SCPs involved in different sulfur transfer reactions, as is known from other organisms such as $Haloferax$ $volcanii$[21] and $Thermococcus$ $kodakarensis$[23] that use several SCPs. To conclude, our gene shuffle approach has uncovered that the conserved Urm1-like conjugation function of $Saci$Urm1 can be separated in yeast from the tRNA thiolation function of $Sc$Urm1.

## Conclusions

Our gene shuffle experiments demonstrate that stable expression of an archaeal Urm1 homolog ($Saci$Urm1) in yeast can pave the way for evolutionary studies on prokaryotic and eukaryotic members of the ubiquitin-like protein family. We show $Saci$Urm1 cooperates with yeast Urm1 pathway components, in particular Uba4, for thioactivation and engages in sulfur-dependent Urm1-like conjugation to Ahp1. The latter peroxiredoxin is a bona fide Urm1 target in yeast, indicating that ubiquitin-like urmylation is conserved among prokaryotes and eukaryotes. Despite detectable thioactivation in vivo, $Saci$Urm1 fails to support sulfur transfer for tRNA thiolation in yeast. Hence, dual-functional sulfur carrier roles for Urm1 in tRNA thiolation and protein urmylation can be separated from one another by expressing an archaeal Urm1 homolog in yeast. Such scenario may be beneficial for further research that focuses on urmylation mechanisms and relevance, particularly when tRNA thiolation is to be avoided. Alternatively, in an attempt to rescue the tRNA thiolation incompatibility and overcome cross-species barriers, such separation of function scenarios may provide new evolutionary insights into Urm1 dual-functionality. Hence, we consider the Urm1-like modifier

from $Sulfolobus$ at a phylogenetic position suited to further investigate evolutionary roots of the ubiquitin-like protein family.

## Methods

### Yeast strains and plasmid constructions

$S.$ $cerevisiae$ strains used in this study (Supplementary Table 1) were grown on complete (YPD) or synthetic (SC) media according to standard methods[39]. Thermosensitivity was analyzed on SC medium at 30 °C or 37 °C, while drug sensitivity was analyzed on YPD medium at 30 °C using 0.02 mg/ml zeocin, after 2–3 days. Yeast cell transformations with PCR products or gene shuffle plasmids (Supplementary Table 2) were done according to Gietz & Woods[40]. Genomic deletions were generated by PCR[41,42] with gene-specific oligonucleotides and confirmed with oligonucleotides binding outside the target loci (Supplementary Table 3).

For multi-copy $ScURM1$ expression, pAJ45 (Supplementary Table 2) was amplified with primers M13_FW_-40 and M13_RV_-27 (Supplementary Table 3) to obtain HA-tagged $ScURM1$ with $ADH1$ promoter and $CYC1$ terminator. The amplified product was cut with $Hin$dIII and $Sal$I and ligated into YEplac195, resulting in pKZ25 (Supplementary Table 2). $SaciURM1$ was amplified directly from archaeal genomic DNA with primers SaciUrm1_Fw_NotI and SaciUrm1_Rv_MlsI (Supplementary Table 3) and cloned into pKZ25 (Supplementary Table 2), replacing $ScURM1$ with $SaciURM1$, resulting in pKZ6 (Supplementary Table 2). Similarly, $SaciURM1$ and $ScURM1$ were amplified from pKZ6 and pKZ25 (Supplementary Table 2) with reverse primers SaciURM1dG_MlsI_RV or ScURM1dG_MlsI_RV, deleting the C-terminal glycine residue ($\Delta$G), generating plasmids pKZ48 and pKZ50 (Supplementary Table 2), respectively. For single-copy $SaciURM1$ expression, pKZ6 (Supplementary Table 2) was amplified with primers M13_FW_-40 and M13_RV_-27 (Supplementary Table 3) to obtain $P_{ADH1}$-3xHA-$SaciURM1$-$T_{CYC1}$ that was cloned into YCplac33 via $Hin$dIII and $Sal$I, resulting in pKZ2 (Supplementary Table 2). To purify $Sc$Urm1 and $Saci$Urm1 with the Strep-Tactin®XT 4Flow® high capacity resin (IBA Lifesciences, 2-5030-002), a construct of $P_{ADH1}$-8xHis-TwinStrep-$T_{CYC1}$ was synthesized by GenScript Biotech (Netherlands) B.V. and cloned into YEplac195, resulting in pLK247 (Supplementary Table 2). The HA-tagged $URM1$ genes were amplified from pKZ25 and pKZ6 (Supplementary Table 2) and cloned into pLK247 (Supplementary Table 2) via $Not$I and $Mls$I, resulting in pKZ76 and pKZ77 (Supplementary Table 2), respectively. Truncated $\Delta$G mutants thereof were generated as described above, resulting in plasmids pKZ82 and pKZ83 (Supplementary Table 2).

For expression of FLAG-tagged $UBA4$ and a $C397S$ mutant, the $P_{ADH1}$-$UBA4$-$T_{CYC1}$ construct from pAJ16 (Supplementary Table 2) was cloned via $Bam$HI and $Eco$RI in pRS313 (Supplementary Table 2) and N-terminal FLAG tagging was performed with primers N-terminal FLAG FW and N-terminal FLAG RV (Supplementary Table 3), resulting in pKZ60 (Supplementary Table 2). Site-directed mutagenesis was done with primers UBA4_C397S_FW and UBA4_C397S_RV (Supplementary Table 3) to generate pKZ65 (Supplementary Table 2). All generated plasmids were checked by Sanger sequencing (Microsynth Seqlab GmbH).

### Phylogenetic analysis

Phylogenetic trees of ubiquitin-like proteins from Eukaryota, Archaea and Bacteria were inferred using amino acid sequences obtained from the publicly available UniProt database[43]. Multiple sequence alignment was done by MAFFT version 7 using L-large-INS-1 as tree-based progressive method with default parameters[44,45]. Alignment was improved by removing homoplastic and random-like positions with Noisy (v1.5.12.1) and default parameters[46,47]. Maximum likelihood analyses were conducted using RAxML v8.2.12[48] implemented in raxmlGUI2.0.13[49]. The best tree was obtained from 20 independent runs under empirical LG + G model, which was selected with ModelTest-NG v0.1.7[50] implemented in raxmlGUI. Bootstrap support values were calculated from 100 pseudoreplicates of the alignment. The resulting best tree was visualized by iTol version 6[51] and edited with CorelDRAW Suite X5.

https://doi.org/10.1038/s42003-025-09212-3 **Article**

## Urmylation studies

Protein urmylation analysis used a lysis buffer for mechanical yeast cell disruption (10 mM K-HEPES pH 7.0, 10 mM KCl, 1.5 mM MgCl2, 0.5 mM PMSF, 2 mM benzamidine) containing complete protease inhibitors and 2.5 mM NEM. To detect unconjugated Urm1, NEM was excluded. Samples for Western blot analysis were prepared with equal protein concentrations according to Bradford[52] in SDS sample buffer (62.5 mM Tris-HCl pH 6.8, 2% SDS, 10% glycerol, 0.002% bromophenol blue, 5% β-mercaptoethanol) after Laemmli[53]. Samples were run by SDS-PAGE and proteins transferred to PVDF membranes. For protein detection, primary anti-HA (1:5000; F7, Santa Cruz Biotechnology or 2-2.2.14, Invitrogen) and anti-FLAG (1:5000; FG4R, Invitrogen) antibodies were used. Expression of Ahp1 was verified with anti-Ahp1 serum[54] (1:4000). Protein loading was controlled with anti-Cdc19 antibody (1:400000). Western blot signals were visualized using the LiCor Odyssey FC Imager System and Image Studio software (v.5.2.5, Li-COR Biosciences). All Western blots were performed with a minimum of three biological replicates.

## Confirmation of Urm1 thiocarboxylation

To confirm thiocarboxylation of Urm1 in vivo, an additional Twin-Strep-tag® was used for protein purification using the Strep-Tactin®XT 4Flow® high capacity resin (IBA Lifesciences, 2-5030-002). Yeast cells grown to logarithmic phase were mechanically lysed in wash buffer W (100 mM Tris-HCl, pH 8.0, 150 mM NaCl, 1 mM EDTA) containing complete protease inhibitors and 0.1% (v/v) NP-40. Proteins were purified according to manufacturer's instructions by eluting in buffer BXT (100 mM Tris-HCl, pH 8.0, 150 mM NaCl, 1 mM EDTA, 50 mM biotin). Western blot analysis was performed as described above with slight modifications. Specifically, SDS-PAGE gels were supplemented with 20 μg/ml APM[28], and sample buffer was used without β-mercaptoethanol.

## Yeast tRNA isolation and wobble uridine modification profiling

Yeast tRNA was isolated as described[55], using NucleoZOL reagent (Macherey-Nagel). The pelleted tRNA was washed once with 75% ethanol and stored in 100% ethanol. LC-MS/MS analysis of wobble uridine modifications ($ncm^5U_{34}$, $mcm^5U_{34}$, $mcm^5s^2U_{34}$) was performed according to Biedenbänder et al[56]. with the analyzed amount increased to 1 μg of digested RNA sample spiked with 100 ng of internal standard. Absolute quantification of biological triplicates ($n = 3$) used internal and external calibration as described[57] with the total amount of modified nucleosides being normalized to the amount of uridines.

## Statistics and reproducibility

All Western blots and phenotypic growth tests were performed with a minimum of three biological replicates. Graphical illustration and statistical analysis were performed with GraphPad Prism (version 8.0.2). LC-MS/MS data in Fig. 5A represent the mean ± SD from $n = 3$ biologically independent samples and individual data points are displayed as dots. Data values are available in Supplementary Table 4 and the Supplementary Data Excel file. Statistics were performed with two-sided Student's $t$-tests with $p$-values < 0.005 indicated by ** and non-significant changes with ns.

## Reporting summary

Further information on research design is available in the Nature Portfolio Reporting Summary linked to this article.

## Data availability

Complete data for Fig. 5A is available in Supplementary Table 4 and the Supplementary Data Excel file. Full Western blot and phenotypic growth test images are shown in the Supplementary Raw Data section of the Supplementary Information file. The Multiple sequence alignment for phylogenetic analysis can be provided in FASTA format upon reasonable request. All other data are incorporated into the article and its Supplementary Information as well as the Supplementary Data Excel file.

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

## Acknowledgements

The authors thank Dr Kuge (Tohoku Pharmaceutical University, Japan) and Dr Thorner (University of California, USA) for providing anti-Ahp1 and anti-Cdc19 antibodies, respectively, and Dr Ohsumi (Tokyo Institute of Technology, Japan) and Dr Robinson (University of Lancaster, United Kingdom) for the HA-Urm1 expression plasmid and *SaciURM1* DNA, respectively. We also thank Drs Igloi and Albers (University of Freiburg, Germany) for kindly donating APM and reading the manuscript and Drs West (The College of Wooster, USA), Hering, Mayer, and Klassen, as well as Niklas Metzendorf

(University of Kassel, Germany) for their technical assistance and valuable comments on the manuscript. We acknowledge Urm1 project support by Deutsche Forschungsgemeinschaft (DFG), Germany, to R.S. (SCHA750/15-2) and their SPP1784 program *Chemical Biology of Native Nucleic Acid Modifications* to R.S. (SCHA750/20-2). Work in the laboratory of M.H. was supported by DFG Collaborative Research Center *RMaP* (TRR 319, TP C03, Project Id 439669440). L.K. has been awarded a PhD scholarship by the Otto Braun-Fonds (B. Braun, Melsungen AG, Germany). L.K. and K.Z. received support by ZFF-Pilot grants #2887 and #2620 (Zentraler Forschungsfonds, University of Kassel, Germany) awarded to R.S.

## Author contributions

K.Z., L.K., and R.S. designed the experiments; K.Z. and L.K. performed the phylogenetic analyses and the molecular biological experiments; L.B. carried out the LC-MS/MS; K.Z., L.K., L.B., M.H., and R.S. analyzed the data; K.Z., L.K., and R.S. wrote the paper; M.H. and R.S. secured funding support (for details, see above).

## Funding

## Competing interests

Author M.H. declares the following competing interest: consultant for Moderna Inc. All other authors declare no competing interests.
