## [Transparent Peer Review file · Communications Biology]

Evolutionary Conservation of Ubiquitin-like Protein Urmylation as Revealed by URM1 Gene Shuffle from Archaea to Yeast

Corresponding Author: Professor Raffael Schaffrath

Version 0:

Reviewer comments:

Reviewer #1

(Remarks to the Author)

Eukaryotic ubiquitin-like protein modifiers are evolutionarily related to prokaryotic sulphur-carrier proteins. URM1, found in both eukaryotes and prokaryotes, is an interesting molecular “fossil” that has both sulphur-carrier and protein modifier functions. The elegant and rigorous study by Zupfer*, Kadhur* et al. describes experiments performed in yeast in which an endogenous yeast URM1 gene was replaced with archaeal URM1. The ectopically expressed archaeal orthologue was functional in protein modification, but not in the sulphur-carrier function (thiolation of tRNAs), offering a separation-of-function situation that can be useful for further studies. The conclusions are supported by an elegant set of experiments with necessary controls, and complemented with a phylogenetic analysis. Moreover, the study is well-written and illustrated. Overall, I think this study can be accepted for publication with minor improvements.

Comments about terminology

I am not fully convinced about the term “gene shuffling” used in the title and throughout the text. While it seems technically correct (an archaeal gene was introduced into yeast, so the resultant strain is a “shuffled” genetic hybrid), I think it can be confusing. The most common usage of “gene shuffling” suggests random mixing or recombination of genes or, perhaps even preferentially, of gene fragments, to produce complex, shuffled mixes of DNA from two (or more) origins. What the authors did could, in my view, be better described as simply ectopically expressing an archaeal URM1 gene in yeast, or something similar (also, could the authors clarify in the Results if it is transient transfection or stable integration?). I would suggest changing the terminology in the title and the text to something other than “gene shuffling” and perhaps using this term only in the Discussion and with clarification. But I leave it for the authors and editor to decide. I understand that terminology preferences can be sensitive.

Another term that, to my taste, was used slightly outside its standard usage was EMSA. While again it seems technically correct, because a band is shifting upwards, EMSA is almost always in my experience used to refer to studying noncovalent interactions (and especially between proteins and nucleic acids), where a band shifts due to retardation upon noncovalently associating with a partner. Here, the authors are investigating covalent modification using denaturing, reducing SDS-PAGE. I would suggest changing the term EMSA to SDS-PAGE-based analysis of urmylation through band shift. But again, this is a question of taste and I leave it to authors and editor to decide.

Lastly, URM1 gene/protein is sometimes written “URM1” and sometimes “Urm1” – is there a logic to it? Probably yes, but please check.

Specific comments

Line 68: The authors mention di-glycine motif as a constant feature of Ub-like proteins. However, some proteins of this type, including UFM1 and I think also ATG8-family proteins, have only a single C-terminal Gly that is nonetheless functional in conjugation to substrates. Perhaps this could be mentioned. Or just write that they have a terminal di-glycine or more rarely single glycine. It could also be mentioned that in some Ub-like proteins (e.g. SUMO) this glycine must be exposed through proteolytic maturation.

Line 75: The term thiocarboxylate might be slightly confusing, as it can technically refer to a thione or a thiol form. This becomes clear with a schematic, but this is buried in the supplements. Perhaps the authors could consider moving Supplementary Figure 1 to the main text, as it seems to me to be really important to understand the introduction and the study as such. The authors could also consider adding a panel that compares the Urm1 cascade to that of ubiquitin, to highlight mechanistic differences. But this is just a suggestion.

Lines 114 and 115: The authors refer to eukaryotic Ubl proteins as monophyletic and SCPs as paraphyletic. But is this with

respect to the organisms in which these proteins are found or the proteins (genes) themselves? Maybe I am confused, but to me MoaD, ThiS, and MOCS2A comprise descendants of one ancestral protein and could therefore be said to be monophyletic, even if organisms in which they are found (some prokaryotic, some eukaryotic) are then paraphyletic. Again, I might be confused because I am not an expert in protein evolution, but please check.

Line 171: I would delete "either" as the sentence lists three items, not just two.

Fig. 3B: Having SH group within Uba4 and in the thiocarboxylate product in the same red colour can be interpreted as this group being transferred from Uba4 onto Urm1, which I believe is not the case. Perhaps the SH group of Uba4 does not need to be shown or it can be just black. But this is just a suggestion.

Fig. 4A: Is it possible to have some measure of statistical significance of these differences? Also, would it be interesting to provide a representative LC/MS profile?

Line 242: How does tQKE overexpression rescue the lack of tRNA thiolation? Does it mean that these tRNAs are less functional when non-thiolated, but bringing their levels up, even without thiolation, rescues this deficiency? Perhaps you could explain it briefly in the text, as otherwise it is puzzling.

Line 253: When you write that a sulphur donor function of archaeal Urm1 "may still exist in its host organism" this sounds as if it is an unlikely option (but still an option). But I guess that – given SCP is a more ancestral Ubl function – it is actually likely that archaeal Urm1 does have this function, but simply cannot interact with the eukaryotic thiolase complex, as you suggest. If that's the case, maybe you can change "may still exist in its hosts organism" to "likely exists in its host organism". By the way, some "gene shuffling" where fragments of archaeal and yeast URM1 are shuffled to produce hybrid proteins could perhaps be interesting in the future to find a part of URM1 that makes the difference (but this is not a suggestion for the current manuscript revision, just a random idea that came to my mind).

Reviewer #2

(Remarks to the Author)

The present study tests an archaeal homolog of the yeast modifier URM1 for its functionality in yeast, relying on the URM1 activation system of yeast. They find that the *Sulfolobus* URM1 homolog gets activated and can modify AHP1, just like it was known for yeast URM1. However, the archaeal version of URM1 was unable to fulfill the tRNA modification role of scURM.

This is a relatively simple manuscript with a limited scope, which is nevertheless interesting for understanding the evolution of ubiquitin-like modification systems. I don't have any fundamental criticisms, but would like to see a few additional validation data.

- Most data figures just show the results using the archaeal URM1; the text then mentions that the results are identical to those of the authentic yeast protein. Why didn't they show yeast and archaeal results side by side, allowing the reader to better appreciate the similarity or maybe some interesting differences.

- In Figure 1A, the middle panel shows the anti-AHP1 signals at ~20kDa but cuts away the part where the modified protein would run. Thus, the reader cannot judge the portion of modified AHP1 relative to the unmodified form.

- The authors claim that the archaeal Urm1 modifies K32 of AHP1 and show that K32 is important. Why don't they analyze the modification site of AHP1 by GG-remnant MS? In comparison to yeast Urm1. Given that AHP1 is tagged, a GG enrichment would not even be necessary.

- no attempt is made to find out why tRNA modification does not work with archaeal Urm1, it is not even mentioned in the discussion. Some thoughts (or even experiments) would have been helpful here. My first guess is that archaeal Urm1 is not properly recognized by the tRNA modification enzymes.

Reviewer #3

(Remarks to the Author)

In the paper by Zupfer et al., "Evolutionary Conservation of Ubiquitin-like Protein Urmylation as Revealed by URM1 Gene Shuffle from Archaea to Yeast", the authors hypothesize that SaciUrm1 from the thermophilic archaeon *Sulfolobus acidocaldarius* is placed at the crossroads of prokaryotic sulfur transfer and eukaryotic protein conjugation pathways, and they investigate ubiquitin-like urmylation and sulfur transfer for tRNA thiolation. The logic in the paper is very clear, and the results derived from it are convincing and reliable. The reviewers' comments are as follows.

It is very clear that the authors demonstrated Urm1 functional conservation in *Saccharomyces cerevisiae* using gene shuffle from Saci_0669, a ubiquitin-like protein derived from *S. acidocaldarius*. However, it is unclear why Saci_0669 was initially considered to function in tRNA thiolation. As shown in Supplementary Figure 2, *S. acidocaldarius* possesses three ubiquitin-like proteins (Saci_0669, Saci_0952, and Saci_1652). As there is no evidence that Saci_0669 participates in tRNA thiolation in *S. acidocaldarius*, the conclusion that "SaciUrm1 fails to mediate tRNA thiolation" in *S. cerevisiae* should be considered with caution, as it rests on an unverified premise and thus represents a potential weakness in the study's logical framework.

Saci_0952 is longer (235 aa) than other Ubls and is a fusion protein with a molybdopterin synthase catalytic subunit (MoaE), suggesting a role in molybdopterin biosynthesis. In contrast, Ubls involved in tRNA biosynthesis, such as TtuB in *Thermus thermophilus* (Shigh et al., *EMBO J.*, 27(24):3267-78, 2008), SAMP2 in *Haloflex volcanii* (Miranda et al., *Proc Natl Acad Sci U S A.* 108(11):4417-22, 2011), and Tk1093 in *Thermococcus kodakarensis* (Hidese et al., *mBio* 15:e00534-24, 2024), tend to be relatively short in sequence length. Based on this tendency, it can be hypothesized that Saci_1652 (68 aa) may be required for tRNA thiolation. If it could be demonstrated that Saci_1652 can functionally complement tRNA thiolation in *S. cerevisiae*, this would indicate that *S. cerevisiae* possesses functional plasticity in its associated protein set, thereby

providing a basis for evolutionary considerations regarding the convergence of such functions into Urm1. This point warrants explicit discussion by the authors.

Minor

Line 240: For readers unfamiliar with tRNA modifications, it is not immediately clear from the main text why the loss of U34 thiolation results in thermo-sensitive growth. A careful explanation of this point is necessary.

Typo, line 268, thermo-sensitive growth

Version 1:

Reviewer comments:

Reviewer #1

(Remarks to the Author)

I thank the Authors for the assertive but respectful answers to the points I raised. I learned some new things. I consider all my points satisfied and thank the Authors for this exchange and their interesting study.

Reviewer #2

(Remarks to the Author)

I do not fully agree with the authors that comparing the AHP1 modification site for yeast and archaeal URM1 would be "outside the scope" and would require a study on its own. I consider these to be relatively simple experiments, given that the present study already uses tagged modifiers and identifies the modification products by WB. However, my original comment was more a suggestion than a requirement, so this is fine by me.

All other issues have been addressed to my satisfaction.

Reviewer #3

(Remarks to the Author)

Acceptance is recommended.

Nota bene authors' comment:

All changes or alterations in the text, illustrations or bibliography, which we undertook in response to the points raised by the reviewers below, are comprehensively highlighted in yellow and have been transferred to our revised manuscript (COMMSBIO-25-5761A).

Reviewer #1 (Remarks to the Author):

Eukaryotic ubiquitin-like protein modifiers are evolutionarily related to prokaryotic sulphur-carrier proteins. URM1, found in both eukaryotes and prokaryotes, is an interesting molecular “fossil” that has both sulphur-carrier and protein modifier functions. The elegant and rigorous study by Zupfer, Kadhur* et al. describes experiments performed in yeast in which an endogenous yeast URM1 gene was replaced with archaeal URM1. The ectopically expressed archaeal orthologue was functional in protein modification, but not in the sulphur-carrier function (thiolation of tRNAs), offering a separation-of-function situation that can be useful for further studies. The conclusions are supported by an elegant set of experiments with necessary controls, and complemented with a phylogenetic analysis. Moreover, the study is well-written and illustrated. Overall, I think this study can be accepted for publication with minor improvements.*

Authors' response:

We agree with reviewer #1 that an archaeal Urm1 orthologue operates in ubiquitin-like protein urmylation upon ectopic expression in yeast – but not as a sulphur-carrier (for thiolation of tRNAs) and that our study therefore, offers a separation-of-function situation, which can be useful for further studies into the evolution of ubiquitin-like modification systems. This is essentially one of the major messages in our manuscript for *Communications Biology* and its special collection *Ubiquitin-like Modifications*.

Comments about terminology

I am not fully convinced about the term “gene shuffling” used in the title and throughout the text. While it seems technically correct (an archaeal gene was introduced into yeast, so the resultant strain is a “shuffled” genetic hybrid), I think it can be confusing. The most common usage of “gene shuffling” suggests random mixing or recombination of genes or, perhaps even preferentially, of gene fragments, to produce complex, shuffled mixes of DNA from two (or more) origins. What the authors did could, in my view, be better describes as simply ectopically expressing an archaeal URM1 gene in yeast, or something similar (also, could the authors clarify in the Results if it is transient transfection or stable integration?). I would suggest changing the terminology in the title and the text to something other than “gene shuffling” and perhaps using this term only in the Discussion and with clarification. But I leave it for the authors and editor to decide. I understand that terminology preferences can be sensitive.

Authors' response:

We respect the comment of reviewer #1 who considers the term ‘gene shuffle’ to mean ... “mixing or recombination of genes or, perhaps even preferentially, of gene fragments, to produce complex, shuffled mixes of DNA from two (or more) origins” ... However, as recently put forward in a *Science Advanced* focus article by the group of Evolutionary Biologist Prof William Martin (Raval *et al.* 2023), ‘genes are shuffled’ via horizontal gene transfer (HGT) across genomes and species boundaries over evolutionary time, constantly generating new gene collections in genomes. In essence, our ‘gene shuffle’ experiment represents a form of HGT, too, that (though artificial and man-made) shuffles an archaeal gene into the genomic context of a model eukaryote. Hence, we would like to keep the term ‘gene shuffle’.

Raval PK, Martin WF, Gould SB (2023) Mitochondrial evolution: Gene shuffling, endosymbiosis, and signaling. *Sci Adv* 9: eadj4493. doi: 10.1126/sciadv.adj4493.

We welcome the reviewer's enquiry whether ectopic expression in yeast involved "*transient transfection or stable integration*". The construction of centromeric and/or episomal plasmids used for yeast transformation and stable ectopic *URMI* expression had been originally explained in full detail in the Supplementary Methods section of the Supporting Information. We have now referred to this explanation in the Materials & Methods chapter of the revised script and briefly stated the following sentence:

... "Yeast cell transformations with PCR products or **gene shuffle** plasmids (Supplementary Table 2) were done as previously described⁴⁸. For detailed plasmid construction, see Supplementary **Methods**" ...

Another term that, to my taste, was used slightly outside its standard usage was EMSA. While again it seems technically correct, because a band is shifting upwards, EMSA is almost always in my experience used to refer to studying noncovalent interactions (and especially between proteins and nucleic acids), where a band shifts due to retardation upon noncovalently associating with a partner. Here, the authors are investigating covalent modification using denaturing, reducing SDS-PAGE. I would suggest changing the term EMSA to SDS-PAGE-based analysis of urmylation through band shift. But again, this is a question of taste and I leave it to authors and editor to decide.

Authors' response:

Reviewer #1 questions the term 'electrophoretic mobility shift assay (EMSA)' that we use to indicate a band shift in a gel due to non-covalent adduction of a protein (Urm1) with a thiol-binding agent (i.e., APM: Fig. 4) or formation of a covalent protein (Urm1) conjugate (i.e., Ahp1 urmylation: Fig. 2 & 3). Either way, the Urm1 band that undergoes APM adduction or attachment to Ahp1 will shift in size due to retardation in native or denaturing gels. Hence it distinguishes itself from the free, unmodified form. So technically speaking, EMSA is a term that describes the gel shifts (which are central to our script) correctly. Moreover, since its establishment by Garner & Revzin (1981) for protein and nucleic acid interaction analysis, the EMSA protocol has been further developed to allow to detect band shifts that involve biomolecular modifications including (but not restricted to) phosphorylations and ADP-ribosylations of proteins as well as chemical decorations of tRNAs and underlying interactions with tRNA modifying enzymes (Chramiec-Głąbik et al. 2023).

Garner MM, Revzin A (1981) A gel electrophoresis method for quantifying the binding of proteins to specific DNA regions: application to components of the Escherichia coli lactose operon regulatory system. *Nucleic Acids Res* 9: 3047–3060. doi: 10.1093/nar/9.13.3047.

Chramiec-Głąbik A, Rawski M, Glatt S, Lin T-Y (2023) Electrophoretic mobility shift assay (EMSA) and microscale thermophoresis (MST) methods to measure interactions between tRNAs and their modifying enzymes. *Methods Mol Biol* 2666: 29–53. doi: 10.1007/978-1-0716-3191-1_3.

We would therefore like to keep the term "EMSA" rather than use the bulky term "*SDS-PAGE-based analysis of urmylation through band shift*".

Lastly, URMI gene/protein is sometimes written "URMI" and sometimes "Urm1" – is there a logic to it? Probably yes, but please check.

Authors' response:

Yes, there is a logic to it: according to the standard *Saccharomyces* nomenclature (<https://www.yeastgenome.org>), '*URMI*' (capital letters in italics) refers to the structural gene while '*Urm1*' (no italics) denotes the protein product encoded by the *URMI* gene.

Specific comments

Line 68: The authors mention di-glycine motif as a constant feature of Ub-like proteins. However, some proteins of this type, including UFM1 and I think also ATG8-family proteins, have only a single C-terminal Gly that is nonetheless functional in conjugation to substrates. Perhaps this could be mentioned. Or just write that they have a terminal di-glycine or more rarely single glycine. It could also be mentioned that in some Ub-like proteins (e.g. SUMO) this glycine must be exposed through proteolytic maturation.

Authors' response:

In agreement with reviewer #1, we adjusted the description of the di-glycine motif as a constant feature. The new text reads as follows:

... “Characteristic structural features of Ubl members are conserved throughout evolution and include a β -grasp fold (β -GF), composed of five β -sheets with a central α -helix, and a C-terminal di-glycine motif or more rarely, a single glycine motif (Fig. 1A, B)⁷. Some Ubl members (e.g. SUMO) are even expressed in inactive precursor forms, where the glycine motif must be exposed through proteolytic maturation⁸.” ...

Line 75: The term thiocarboxylate might be slightly confusing, as it can technically refer to a thione or a thiol form. This becomes clear with a schematic, but this is buried in the supplements. Perhaps the authors could consider moving Supplementary Figure 1 to the main text, as it seems to me to be really important to understand the introduction and the study as such. The authors could also consider adding a panel that compares the Urm1 cascade to that of ubiquitin, to highlight mechanistic differences. But this is just a suggestion.

Authors' response:

We thank reviewer #1 for this suggestion and moved an adjusted excerpt from Supplementary Fig. 1 as a schematic into part A (see below) of new main Fig. 3.

New Fig. 3:
The newly designed part A now shows upstream sulfur mobilization and transfer from Nfs1/Tum1 onto the catalytic cysteine (Cys-397) of the rhodanese domain in Uba4 (for details, see above Fig. 3A). The latter is required for subsequent thioactivation of Urm1-COOH yielding the critical thiocarboxylate, Urm1-COSH. We consider these changes will clarify the reviewer's concern and make the 'thiocarboxylate' issue easier to understand for the readership. Moreover, the new schematic is well suited to lead – side-by-side – into the sulfur-transfer mutants (for details, see below Fig. 3B), which we used to analyse for the sulfur-dependence of Ahp1 urmylation with the archaeal Urm1 homologue (SaciUrm1).

Lines 114 and 115: The authors refer to eukaryotic Ubl proteins as monophyletic and SCPs as paraphyletic. But is this with respect to the organisms in which these proteins are found or the proteins (genes) themselves? Maybe I am confused, but to me Moad, ThiS, and MOCS2A comprise descendants of one ancestral protein and could therefore be said to be monophyletic, even if organisms in which they are found (some prokaryotic, some eukaryotic) are then paraphyletic. Again, I might be confused because I am not an expert in protein evolution, but please check.

Authors' response:

Our analysis was conducted to clarify the phylogenetic relationship between ubiquitin-like beta-GF proteins rather than the organisms, in which these proteins are found. Hence the terms 'monophyletic' and 'paraphyletic' indeed refer to proteins or protein groups. Based on the data set, the rooted tree (Supplementary Fig. 3) clearly indicates paraphyletic sulfur carrier families, since ThiS-like proteins obviously do not share a common ancestor as they occur twice in the tree with different sister group relationships. Furthermore, combining ThiS, Moad and MOCS2A to one „family“ creates a paraphyletic group of proteins, since not all descendants of a common ancestor would be included (i.e., Urm1 and the proteins on the branches between Urm1 and Moad would be excluded). Indeed, the branches of Moad, ThiS and MOCS2A, which cluster below the Urm1 group (Supplementary Fig. 3), are monophyletic but this clade does not comprise all ThiS sequences in the data set as already described. The unrooted tree (see Fig. 1C) displays an unresolved Moad/ThiS/MOCS2A cluster with a trichotome-multifurcated split. To our mind, and in contrast to the view of reviewer #1 (see above), the latter unrooted feature hardly allows for assumptions about ancestry or one ancestral protein !

Line 171: I would delete “either” as the sentence lists three items, not just two.

Authors' response:

Changed as requested by reviewer #1.

Fig. 3B: Having SH group within Uba4 and in the thiocarboxylate product in the same red colour can be interpreted as this group being transferred from Uba4 onto Urm1, which I believe is not the case. Perhaps the SH group of Uba4 does not need to be shown or it can be just black. But this is just a suggestion.

Authors' response:

We chose the same red colour for the SH group within Uba4 and in the Urm1 thiocarboxylate because the persulfide (SH group) of Uba4 is necessary for Urm1 thioactivation. The first interpretation of reviewer #1 that the SH group (of the persulfidated) Uba4 is transferred onto Urm1 is right. The catalytically active cysteine (i.e., Cys-397) in the rhodanese domain of Uba4 is now more precisely depicted in the new Fig. 3A, which we designed to address the related 'thiocarboxylate' enquiry of reviewer #1 (see our response above). We felt this new schematic is helpful for clarifying the 'SH group' issue of reviewer #1 and also added into the legend of the new Fig. 4A: ... **“Urm1-COOH is thio-activated at its C-terminal glycine by Uba4 resulting in Urm1-COSH.”** ...

Fig. 4A: Is it possible to have some measure of statistical significance of these differences? Also, would it be interesting to provide a representative LC/MS profile?

Authors' response:

Thanks to the helpful suggestion of reviewer #1, we now added the statistical significance with p-values into part A of the new revised Fig. 5 (see below for details). The values were determined with Student's *t*-tests that are based on three independent LC-MS/MS

measurements listed in Supplementary Table 4 of the Supporting Information. Also, we added individual data points to the graph and according to the editorial policies, enclosed the data set in Excel format. In our opinion, the data now are clearly presented so that we politely disagree with reviewer #1 that “*provision of a representative LC/MS profile may be interesting*”. In the Materials and Methods section of the revised script, we have referred to the statistics by adding the following:

... “*Graphical illustration and statistical analysis were performed with Graphpad Prism (version 8.0.2). LC-MS/MS data in Fig. 5A represent the mean \pm SD from three biological replicates and individual data points are displayed as dots. Statistics were performed with two-sided Student's t-tests with p-values $<0,005$ indicated by ** and non-significant changes with ns.*” ...

New Fig. 5:

Line 242: How does tQKE overexpression rescue the lack of tRNA thiolation? Does it mean that these tRNAs are less functional when non-thiolated, but bringing their levels up, even without thiolation, rescues this deficiency? Perhaps you could explain it briefly in the text, as otherwise it is puzzling.

Authors' response:

In response to this valuable comment of reviewer #1, which is thematically related to a similar one by reviewer #3 (see below), we added the following text into the revised manuscript:

... “*We also confirmed this defect in vivo by showing that *SaciUrm1* expression cannot rescue phenotypes that result from U_{34} thiolation loss in *urm1Δ* cells (Fig. 5B). In yeast, lack of tRNA thiolation at the wobble position U_{34} does not result in a reduced abundance of hypomodified tRNAs, but has been shown to reduce the binding affinity of the hypomodified anticodon to the cognate codon in the ribosomal A-site^{40,41}. The defect results in inefficient decoding and error-prone mRNA translation, which is why the growth and viability of *URM1* gene deletion (*urm1Δ*) strains becomes significantly compromised, particularly upon cultivation at elevated temperatures or in the presence of chemical stressors such as zeocin (Fig. 5B). Intriguingly, overexpression of hypothiolated tRNAs (i.e., *tQKE*) compensates for the decoding inefficiency of *urm1Δ* cells and partially rescues the thermo and drug sensitivity phenotypes (Fig. 5B). Unlike with *ScUrm1* or partial rescue by tRNA overexpression (*tQKE*, see above), *SaciUrm1* expression could not complement zeocin sensitivity or synthetic lethality in tandem with tRNA pseudouridylation mutant *deg1Δ* (Fig. 5B)^{13,20,42}” ...*

Johansson MJ, Esberg A, Huang B, Björk GR, Byström AS (2008) Eukaryotic wobble uridine modifications promote a functionally redundant decoding system. *Mol Cell Biol* **28**: 3301-3312. doi: 10.1128/MCB.01542-07.

Ranjan N, Rodnina MV (2017) Thio-Modification of tRNA at the Wobble Position as Regulator of the Kinetics of Decoding and Translocation on the Ribosome. *J Am Chem Soc* **139**: 5857-5864. doi: 10.1021/jacs.7b00727.

In our opinion, these additional explanations on the consequences of tRNA overexpression and loss of tRNA thiolation have clarified the points raised by both reviewer #1 and #3 (see below).

Line 253: When you write that a sulphur donor function of archaeal Urm1 “may still exist in its host organism” this sounds as if it is an unlikely option (but still an option). But I guess that – given SCP is a more ancestral Ubl function – it is actually likely that archaeal Urm1 does have this function, but simply cannot interact with the eukaryotic thiolase complex, as you suggest. If that’s the case, maybe you can change “may still exist in its hosts organism” to “likely exists in its host organism”.

Authors’ response:

We agree with reviewer #1 that the proposed wording “*likely exists in its host organism*” suits our hypothesis better. Accordingly, we have reworded the sentence in our revised script.

By the way, some “gene shuffling” were fragments of archaeal and yeast URM1 are shuffled to produce hybrid proteins could perhaps be interesting in the future to find a part of URM1 that makes the difference (but this is not a suggestion for the current manuscript revision, just a random idea that came to my mind).

Authors’ response:

The idea to use ‘gene shuffle’ experiments for future Urm1 dissection and functional analysis is very intriguing and we do thank reviewer #1 for sharing it with us.

Reviewer #2 (Remarks to the Author):

The present study tests an archaeal homolog of the yeast modifier URM1 for its functionality in yeast, relying on the URM1 activation system of yeast. They find that the Sulfolobus URM1 homolog gets activated and can modify AHPI, just like it was known for yeast URM1. However, the archaeal version of URM1 was unable to fulfill the tRNA modification role of scURM.

This is a relative simple manuscript with a limited scope, which is nevertheless interesting for understanding the evolution of ubiquitin-like modification systems. I don't have any fundamental criticisms, but would like to see a few additional validation data.

Authors’ response:

We agree with reviewer #2 that our study, though relatively simple, offers to understand the evolution of ubiquitin-like modification systems. This was on purpose and exactly the major scope of our manuscript in format of a report for the collection *Ubiquitin-like Modifications*.

- Most data figures just show the results using the archaeal URM1; the text then mentions that the results are identical to those of the authentic yeast protein. Why didn't they show yeast and archaeal results side by side, allowing the reader to better appreciated the similarity or maybe some interesting differences.

Authors' response:

Following the helpful suggestion of reviewer #2, we extended the data-sets in new Fig. 2 and new Fig. 4 (for details, see below). Shown are side-by-side *Sulfolobus* and *Saccharomyces* presentations to indeed highlight the striking similarities between both the archaeal and yeast urmylation results.

New Fig. 2 (part A and B only):**New Fig. 4 (part B and C only):**
Regarding other urmylation aspects (that inspired our analysis and are also covered in our script), there are several in-depth studies on the yeast urmylation pathway and its requirements. These have been published in papers (Jüdes *et al.* 2016; Brachmann *et al.* 2020; Ravichandran *et al.* 2022) that we already cited in the relevant sections of our original submission. Therefore, we kindly refrain from showing more side-by-side comparisons, for this would simply result in a duplication of native yeast urmylation that are already published:

Jüdes A, Bruch A, Klassen R, Helm M, Schaffrath R (2016) Sulfur transfer and activation by ubiquitin-like modifier system Uba4•Urm1 link protein urmylation and tRNA thiolation in yeast. *Microb Cell* **3**: 554-564. doi: 10.15698/mic2016.11.539.

Brachmann C, Kaduhr L, Jüdes A, Keerthiraju ER, West JD, Glatt S, Schaffrath R (2020) Redox requirements for ubiquitin-like urmylation of Ahp1, a 2-Cys peroxiredoxin from yeast. *Redox Biol* **30**: 101438. doi: 10.1016/j.redox.2020.101438.

Ravichandran KE, Kaduhr L, Skupieñ-Rabian B, Shvetsova E, Sokołowski M, Krutyhołowa R, Kwasana D, Brachmann C, Lin S, Perez SG, Wilk P, Kösters M, Grudnik P, Jankowska U, Leidel SA, Schaffrath R, Glatt S (2022) E2/E3 independent ubiquitin-like conjugation by Urm1 is directly coupled to cysteine persulfidation. *EMBO J* **41**: e111318. doi: 10.15252/embj.2022111318.

To exemplify, we like to show to the reviewer's eyes some of these yeast urmylation data, which we intend to avoid duplicating:

Brachmann et al. (2020) – Fig. 6C:

Fig. 6C shows different Ahp1 substitution mutants with regard to urmylation by yeast Urm1 under reducing (left panels) or non-reducing (right panels) conditions. In particular, the resolving (C31S) and peroxidatic cysteine (C62S) mutants were of interest: the C31S substitution forms Urm1 conjugates in contrast to C62S and the double C31,62S mutant, which both failed in urmylation. These three *ahp1* mutants (C31S; C62S; C31,62S) were also examined in our script for *Communications Biology* (see Fig. 2C) for their conjugation capacity with *SaciUrm1*, the archaeal homologue of yeast Urm1.

Brachmann et al. (2020) – Fig. 7B:

Fig. 7B shows an analysis of lysine-based acceptor sites in Ahp1 for yeast urmylation. An *ahp1* substitution mutant (K32R) shows significantly decreased urmylation acceptor activity. For this reason, we also examined the K32R mutant for urmylation capacity with *SaciUrm1*, the archaeal homologue of yeast Urm1, in our study addressed to *Communications Biology* (see Fig. 2C).

- In Figure 1A, the middle panel shows the anti-AHP1 signals at ~20kDa but cuts away the part where the modified protein would run. Thus, the reader cannot judge the portion of modified AHP1 relative the the unmodified form.

Authors' response:

We agree with reviewer #2 that immunological detection with the anti-Ahp1 antibody of Urm1-conjugated Ahp1 would be interesting. Unfortunately, as stated in previous studies (Jüdes *et al.*

2016; Brachmann *et al.* 2020; Ravichandran *et al.* 2022), the anti-Ahp1 antiserum (kindly provided by Professor Kuge, Tohoku Pharmaceutical University, Japan) only recognizes unmodified Ahp1 with the critical epitope likely being masked in Urm1-conjugates. Thus, the anti-Ahp1 antibody serves us as a control for Ahp1 expression, not Ahp1 urmylation. In our hands, urmylation is best demonstrated by detecting HA-tagged Urm1 attached to Ahp1 in anti-HA Western blots. For the eye of the reviewer and the sake of completeness, our Supporting Information (see below) presents the raw data for the uncropped anti-Ahp1 blot, in which higher molecular bands typical of urmylated Ahp1 are missing. Because of the detection problematic above, we have to admit that the portion between free and urmylated Ahp1 is not possible to determine using Professor Kuge's anti-Ahp1 antibody, unfortunately.

Supplementary Raw Data:

Raw data for Fig. 2A

Blot for Fig. 2A: anti-Ahp1

- The authors claim that the archaeal Urm1 modifies K32 of AHP1 and show that K32 is important. Why don't they analyze the modification site of AHP1 by GG-remnant MS? In comparison to yeast Urm1. Given that AHP1 is tagged, a GG enrichment would not even be necessary.

Authors' response:

We appreciate this suggestion of reviewer #2 and consider it is particularly intriguing for in-depth structure/function analysis of the Urm1 conjugation sites in the target protein Ahp1. However, this was by no means the scope of our work, which aimed at elaborating conserved principles in ubiquitin-like urmylation throughout evolution. To us, it is self-evident that a thorough structure/function analysis of Ahp1 conjugation sites for the archaeal Urm1 homologue would necessitate a separate exercise/study. Furthermore, ground-breaking aspects with regards to lysin-directed Urm1 acceptor activity have already been revealed by cryo-EM and MS analysis of Ahp1 from *Saccharomyces cerevisiae* and *Chaetomium thermophilum* (Ravichandran *et al.* 2022). Their study uncovered that K32 is not the only and unique target site for urmylation with yeast Urm1. In the K32R mutant, the level of urmylation is significantly reduced but not completely abolished, a scenario we also observe in our script with residual urmylation of the K32R mutant by *SaciUrm1*, the archaeal homologue of yeast Urm1 (see Fig. 2C). In order to map urmylation sites other than K32, further lysine residues were mutated by Ravichandran *et al.* (2022) but they could not identify a preferred acceptor site. More strikingly, even in the absence of all Ahp1 surface-exposed lysine residues, residual Urm1 conjugation was identified through mass spectrometry analysis similar to GG-remnant MS at serine and threonine residues of Ahp1 !

Ravichandran KE, Kaduhr L, Skupieñ-Rabian B, Shvetsova E, Sokołowski M, Krutyhołowa R, Kwasana D, Brachmann C, Lin S, Perez SG, Wilk P, Kösters M, Grudnik P, Jankowska U, Leidel SA, Schaffrath R, Glatt S (2022) E2/E3 independent ubiquitin-like conjugation by Urm1 is directly coupled to cysteine persulfidation. *EMBO J* **41**: e111318. doi: 10.15252/embj.2022111318.

- no attempt is made to find out why tRNA modification does not work with archaeal Urm1, it is not even mentioned in the discussion. Some thoughts (or even experiments) would have been helpful here. My first guess is that archaeal Urm1 is not properly recognized by the tRNA modification enzymes.

Authors' response:

We are surprised to find reviewer #2 states we would not have undertaken attempts to discuss why (other than ubiquitin-like urmylation) archaeal Urm1 is not able to support the tRNA thiolation pathway in yeast. In fact, in our original script we suggest this lack of function may result from incompatibility between the archaeal Urm1 homologue and the eukaryotic tRNA thiolation enzyme (Ncs2-Ncs6, see Supplementary Fig S1), which is distinct in subunit structure and stoichiometry from archaeal counterparts:

... "Hence, our data show that despite being thiocarboxylated by Uba4 for sulfur-dependent Urm1-like conjugation (Fig. 4), SaciUrm1 cannot transfer the sulfur for tRNA thiolation in yeast (Fig. 5). Such incompatibility may be explained by SaciUrm1 failing to interact properly with the thiolase complex in yeast. In contrast to eukaryotic thiolase heterodimers, archaeal homologues rather form homodimers, indicating an evolutionary diversification from homo- to heterodimeric tRNA thiolation enzymes^{38,43,44}. Such conflict could reflect an evolutionary barrier that may have resulted in functional specification of the eukaryotic thiolase complex that is unable to recognize and use the archaeal Urm1 protein as sulfur donor^{44,45}" ...

In as such, the thoughts we did articulate in our original manuscript (see above), are indeed helpful and very similar to the above guess of reviewer #2 that ... *"archaeal Urm1 is not properly recognized by the tRNA modification enzymes"* ...

Reviewer #3 (Remarks to the Author):

In the paper by Zupfer et al., "Evolutionary Conservation of Ubiquitin-like Protein Urmylation as Revealed by URM1 Gene Shuffle from Archaea to Yeast", the authors hypothesize that SaciUrm1 from the thermophilic archaeon Sulfolobus acidocaldarius is placed at the crossroads of prokaryotic sulfur transfer and eukaryotic protein conjugation pathways, and they investigate ubiquitin-like urmylation and sulfur transfer for tRNA thiolation. The logic in the paper is very clear, and the results derived from it are convincing and reliable. The reviewers' comments are as follows.

Authors' response:

We appreciate the view of reviewer #3 that the logic in our study is clear with convincing results to hypothesize that archaeal Urm1 is at the crossroads of prokaryotic sulfur transfer and eukaryotic conjugation pathways. This is exactly why we chose our script for consideration of publication by *Communications Biology* (Collection: *Ubiquitin-like Modifications*).

It is very clear that the authors demonstrated Urm1 functional conservation in Saccharomyces cerevisiae using gene shuffle from Saci_0669, a ubiquitin-like protein derived from S. acidocaldarius. However, it is unclear why Saci_0669 was initially considered to function in tRNA thiolation. As shown in Supplementary Figure 2, S. acidocaldarius possesses three ubiquitin-like proteins (Saci_0669, Saci_0952, and Saci_1652). As there is no evidence that Saci_0669 participates in tRNA thiolation in S. acidocaldarius, the conclusion that "SaciUrm1 fails to mediate tRNA thiolation" in S. cerevisiae should be considered with

caution, as it rests on an unverified premise and thus represents a potential weakness in the study's logical framework.

Authors' response:

We think our motivation to start with Saci_0669 and check whether (or not) it is involved in tRNA thiolation had been thoughtfully delineated. As stated in the introduction of our original script (lines 87-93, see below in italics), Saci_0669 was described as an archaeal Urm1 homologue of yeast (Anjum *et al.* 2015), and Urm1 from yeast, plant and human cells has bifunctional roles for eukaryotes in protein urmylation and tRNA thiolation. To improve the illustration of our initial motivation we have added the insert shown below into our introduction part:

*... "Sulfur-dependence of Urm1 functions in tRNA thiolation and urmylation is conserved in eukaryotes^{19,20}, and proteins related to Urm1 have also been identified in Archaea²¹⁻²³. One such Urm1-like protein from *Sulfolobus acidocaldarius* (Saci_0669) was shown to conjugate to proteins depending on an E1-like enzyme²². However, whether this modifier also needs sulfur activation, or operates in sulfur transfer, **such as tRNA thiolation**, has yet to be demonstrated and could provide valuable insights into the evolution of prokaryotic and eukaryotic members of the Ubl protein family." ...*

So, we asked whether Saci_0669 also functions in both of these pathways – when shuffled into the genetic context of a yeast cell. To further base our motivation and convincingly address the question as to whether Saci_0669 is a *bona fide* homologue of yeast Urm1, we carefully conducted phylogenetic analyses (Fig. 1C; Supplementary Fig. 3). The trees obtained showed unambiguously that among three ubiquitin-like proteins predicted to be encoded from open reading frames in the genome of *S. acidocaldarius* (Saci_0669; Saci_0952; Saci_1652), it is solely Saci_0669, which exclusively clusters into the monophyletic Urm1 clade. Furthermore, we found that Saci_0669 is sulfur activated (thiocarboxylated) in yeast (new Fig. 4B, see above for details), a step critical to the sulfur transfer required for tRNA thiolation at U34. To our minds, these findings highlight the potential for Saci_0669 to function in tRNA thiolation similar to eukaryotic Urm1. Thus, we consider the scenario sufficiently evident to choose Saci_0669 in the first place as a potential sulfur carrier protein for tRNA thiolation. We are aware that to this end, there is no data linking Saci_0669 directly to sulfur transfer associated with tRNA thiolation in its *Sulfolobus* host organism. Therefore, we included the following sentence (marked in italics and yellow below) in order to acknowledge the suggestion of reviewer #3 that our conclusion should be considered with caution:

*... "Such conflict could reflect an evolutionary barrier that may have resulted in functional specification of the eukaryotic thiolase complex that is unable to recognize and use the archaeal Urm1 protein as sulfur donor^{44,45}. **To the best of our knowledge there is no data linking SaciUrm1 directly to sulfur transfer associated with tRNA thiolation in *Sulfolobus*. However, archaeal tRNA thiomodifications do exist as has been recently uncovered by comprehensive LC-MS/MS profiles on tRNAs isolated from several Archaea including *S. acidocaldarius*⁴⁶. Thus, a sulfur donor function of SaciUrm1 for tRNA thiolation likely exists in its host organism *S. acidocalarius* and needs further investigations.**" ...*

Wolff P, Villette C, Zumsteg J, Heintz D, Antoine L, Chane-Woon-Ming B, Droogmans L, Grosjean H, Westhof E (2020) Comparative patterns of modified nucleotides in individual tRNA species from a mesophilic and two thermophilic archaea. *RNA* 26: 1957-1975. doi: 10.1261/rna.077537.120.

Saci_0952 is longer (235 aa) than other Ubls and is a fusion protein with a molybdopterin synthase catalytic subunit (MoaE), suggesting a role in molybdopterin biosynthesis. In

contrast, Ubls involved in tRNA biosynthesis, such as *TtuB* in *Thermus thermophilus* (Shigh et al., *EMBO J.*, 27(24):3267-78, 2008), *SAMP2* in *Haloferax volcanii* (Miranda et al., *Proc Natl Acad Sci U S A.* 108(11):4417-22, 2011), and *Tk1093* in *Thermococcus kodakarensis* (Hidese et al., *mBio* 15:e00534-24, 2024), tend to be relatively short in sequence length. Based on this tendency, it can be hypothesized that *Saci_1652* (68 aa) may be required for tRNA thiolation. If it could be demonstrated that *Saci_1652* can functionally complement tRNA thiolation in *S. cerevisiae*, this would indicate that *S. cerevisiae* possesses functional plasticity in its associated protein set, thereby providing a basis for evolutionary considerations regarding the convergence of such functions into *Urm1*. This point warrants explicit discussion by the authors.

Authors' response:

We thank reviewer #3 for this valuable comment and agree that *Saci_1652* would be interesting for future studies regarding sulfur transfer. However, *Saci_1652* was not a central aspect of our study. Our phylogenetic analysis clearly showed that *Saci_0669* was the most relevant of the three *S. acidocaldarius* Ubl proteins (*Saci_0669*; *Saci_0952*; *Saci_1652*), and thus in the focus of our research question as explained in our previous answer. It is evident, to us, that a detailed function analysis of *Saci_1652* would require a separate study. Thus, we believe that the hypothesis raised by reviewer #3, which predicts protein function based solely on sequence length without additional experimental evidence, is highly speculative and does not warrant such explicit discussion at this point. Nevertheless, we acknowledge the importance of this issue raised by reviewer #3 and expanded our discussion to include the recommendation that the two uncharacterised Ubl proteins (*Saci_0952* and *Saci_1652*) require further investigation regarding sulfur transfer. The new text reads as follows:

... “Thus, a sulfur donor function of *SaciUrm1* for tRNA thiolation likely exists in its host organism *S. acidocaldarius* and needs further investigations. These should also include the two open reading frames from *S. acidocaldarius*, predicted to encode Ubl proteins (*Saci_0952* and *Saci_1652*) (Supplementary Fig. 3). They may represent potential SCPs involved in different sulfur transfer reactions, as is known from other organisms, such as *Haloferax volcanii*²¹ and *Thermococcus kodakarensis*²³ that use several SCPs.” ...

Humbard MA, Miranda HV, Lim JM, Krause DJ, Pritz JR, Zhou G, Chen S, Wells L, Maupin-Furlow JA (2010) Ubiquitin-like small archaeal modifier proteins (SAMPs) in *Haloferax volcanii*. *Nature* **463**: 54–60. doi: 10.1038/nature08659.

Hidese R, Ohira T, Sakakibara S, Suzuki T, Shigi N, Fujiwara S (2024) Functional redundancy of ubiquitin-like sulfur-carrier proteins facilitates flexible, efficient sulfur utilization in the primordial archaeon *Thermococcus kodakarensis*. *mBio* **15**: e0053424. doi: 10.1128/mbio.00534-24.

Minor

Line 240: For readers unfamiliar with tRNA modifications, it is not immediately clear from the main text why the loss of U34 thiolation results in thermo-sensitive growth. A careful explanation of this point is necessary.

Authors' response:

In response to the suggestion by reviewer #3, which relates to a similar comment by reviewer #1 (see above), we added the following text into the revised manuscript to improve understanding the consequences of loss of tRNA thiolation and its suppression by tRNA overexpression:

... “We also confirmed this defect in vivo by showing that *SaciUrm1* expression cannot rescue phenotypes that result from U₃₄ thiolation loss in *urm1Δ* cells (Fig. 5B). In yeast, lack of tRNA thiolation at the wobble position U₃₄ does not result in a reduced abundance of hypomodified tRNAs, but has been shown to reduce the

binding affinity of the hypomodified anticodon to the cognate codon in the ribosomal A-site^{40,41}. The defect results in inefficient decoding and error-prone mRNA translation, which is why the growth and viability of *URM1* gene deletion (*urm1Δ*) strains becomes significantly compromised, particularly upon cultivation at elevated temperatures or in the presence of chemical stressors such as zeocin (Fig. 5B). Intriguingly, overexpression of hypothesized tRNAs (i.e., *tQKE*) compensates for the decoding inefficiency of *urm1Δ* cells and partially rescues the thermo and drug sensitivity phenotypes (Fig. 5B). Unlike with *ScUrm1* or partial rescue by tRNA overexpression (*tQKE*, see above), *SaciUrm1* expression could not complement zeocin sensitivity or synthetic lethality in tandem with tRNA pseudouridylation mutant *deg1Δ* (Fig. 5B)^{13,20,42} ...

Johansson MJ, Esberg A, Huang B, Björk GR, Byström AS (2008) Eukaryotic wobble uridine modifications promote a functionally redundant decoding system. *Mol Cell Biol* **28**: 3301-3312. doi: 10.1128/MCB.01542-07.

Ranjan N, Rodnina MV (2017) Thio-Modification of tRNA at the Wobble Position as Regulator of the Kinetics of Decoding and Translocation on the Ribosome. *J Am Chem Soc* **139**: 5857-5864. doi: 10.1021/jacs.7b00727.

Typo, line 268, thermo-sensitive growth

Authors' response:

The error has been corrected to: ... "*thermo-sensitive growth*" ...

REVIEWERS' COMMENTS:

Reviewer #1 (Remarks to the Author):

I thank the Authors for the assertive but respectful answers to the points I raised. I learned some new things. I consider all my points satisfied and thank the Authors for this exchange and their interesting study.

Authors' response:

We also thank reviewer #1 for the scientific in-put and mutual exchange during the reviewing and revision process. We consider this has had a major impact in enhancing our study on ubiquitin-like protein conservation and evolution.

Reviewer #2 (Remarks to the Author):

I do not fully agree with the authors that comparing the AHP1 modification site for yeast and archaeal URM1 would be "outside the scope" and would require a study on its own. I consider these to be relatively simple experiments, given that the present study already uses tagged modifiers and identifies the modification products by WB. However, my original comment was more a suggestion than a requirement, so this is fine by me.

All other issues have been addressed to my satisfaction.

Authors' response:

Although reviewer #2 may not have agreed 100% with us on what to judge an 'outside-the-scope' exercise (or not), we do feel very grateful for most other crucial points raised by the reviewer's criticism. As a consequence of the careful and thorough review process triggered this way, we consider the key message in our study did even more improve, namely that the gene shuffle and complementation strategy used offers to uncover a separation-of-function situation, which can be very instructive for further studies into the evolution of ubiquitin-like modification/urmylation systems. So, reviewer #2 really helped us enhance one of the major aspects in our manuscript for *Communications Biology*.

Reviewer #3 (Remarks to the Author):

Acceptance is recommended.

Authors' response:

Needless to say that the contributions of reviewer #3 (like the constructive criticism of reviewer #1 & reviewer #2) largely improved the quality of our final manuscript. Thank you !